# THE CONVERGENCE RATE OF SGD'S FINAL ITERATE: ANALYSIS ON DIMENSION DEPENDENCE

## ABSTRACT

Stochastic Gradient Descent (SGD) is among the simplest and most popular optimization and machine learning methods. Running SGD with a fixed step size and outputting the final iteration is an ideal strategy one can hope for, but it is still not well-understood even though SGD has been studied extensively for over 70 years. Given the $\Theta(\log T)$ gap between current upper and lower bounds for running SGD for $T$ steps, it was then asked by Koren & Segal (2020) how to characterize the final-iterate convergence of SGD with a fixed step size in the constant dimension setting, i.e., $d = O(1)$. In this paper, we consider the more general setting for any $d \leq T$, proving $\Omega(\log d/\sqrt{T})$ lower bounds for the sub-optimality of the final iterate of SGD in minimizing non-smooth Lipschitz convex functions with standard step sizes. Our results provide the first general dimension-dependent lower bound on the convergence of SGD's final iterate, partially resolving the COLT open question raised by Koren & Segal (2020). Moreover, we present a new method in one dimension based on martingale and Freedman's inequality, which gets the tight $O(1/\sqrt{T})$ upper bound with mild assumptions.

## 1 INTRODUCTION

Stochastic gradient descent (SGD) was first introduced by Robbins & Monro (1951). It soon became one of the most popular tools in applied machine learning, e.g., Johnson & Zhang (2013); Schmidt et al. (2017) due to its simplicity and effectiveness. SGD works by iteratively taking a small step in the opposite direction of an unbiased estimate of sub-gradients and is widely used in minimizing convex function $f$ over a convex domain $\mathcal{K}$. Formally speaking, given a stochastic gradient oracle for an input $x \in \mathcal{K}$, the oracle returns a random vector $\hat{g}$ whose expectation is equal to one of the sub-gradients of $f$ at $x$. Given an initial point $x_1$, SGD generates a sequence of points $x_1, ..., x_{T+1}$ according to the update rule

$$x_{t+1} = \Pi_{\mathcal{K}}(x_t - \eta_t \hat{g}_t) \tag{1}$$

where $\Pi_{\mathcal{K}}$ denotes projection onto $\mathcal{K}$ and $\{\eta_t\}_{t \geq 1}$ is a sequence of step sizes.

Theoretical analysis on SGD usually adopt running average step size, i.e., outputting $\frac{1}{T}\sum_{t=1}^{T} x_t$ in the end, to get optimal rates of convergence in the stochastic approximation setting. Optimal convergence rates have been achieved in both convex and strongly convex settings when averaging of iterates is used Nemirovskij & Yudin (1983); Zinkevich (2003); Kakade & Tewari (2008); Cesa-Bianchi et al. (2004). Nonetheless, the final iterate of SGD, which is often preferred over the running average, as pointed out by Shalev-Shwartz et al. (2011), has not been very well studied from the theoretical perspective, and convergence results for the final iterate are relatively scarce compared with the running average schedule.

**Standard** choices of step sizes for convex functions include $\eta_t = 1/\sqrt{t}$ for unknown horizon $T$ and $\eta_t = 1/\sqrt{T}$ for known $T$, and $\eta_t = 1/t$ for strongly convex functions. In these cases, it is known that the final-iterate convergence rate of SGD is optimal when $f$ is both smooth and strongly convex (Nemirovski et al. (2009)). However, in practice, the convex functions we want to minimize are often non-smooth. See Cohen et al. (2016); Lee et al. (2013) for more details. The convergence rate of SGD's final iterate with standard step sizes in the non-smooth setting is much less explored. Understanding this problem is essential as the final iterate of SGD is popular and used often. If the last iterate of SGD performs as well as the running average, it yields a very simple, implementable,

| Work | Rate | Method | Convexity | Step size | Assumptions |
|---|---|---|---|---|---|
| Nemirovski et al. (2009) | $O(1/T)$ | SGD | Strongly | $1/t$ | Smooth |
| Jain et al. (2019) | $O(1/\sqrt{T})$ | SGD | Convex | Non-standard | |
| Jain et al. (2019) | $O(1/T)$ | SGD | Strongly | Non-standard | |
| Shamir & Zhang (2013) | $O(\log T/\sqrt{T})$ | SGD | Convex | $1/\sqrt{t}$ | |
| Shamir & Zhang (2013) | $O(\log T/T)$ | SGD | Strongly | $1/t$ | |
| Harvey et al. (2019a) | $\Omega(\log T/\sqrt{T})$ | GD | Convex | $1/\sqrt{t}$ | $d \geq T$ |
| Harvey et al. (2019a) | $\Omega(\log T/T)$ | GD | Strongly | $1/t$ | $d \geq T$ |
| Ours | $\Omega(\log d/\sqrt{T})$ | GD | Convex | $1/\sqrt{t}, 1/\sqrt{T}$ | $d \leq T$ |
| Ours | $\Omega(\log d/T)$ | GD | Strongly | $1/t$ | $d \leq T$ |

Table 1: Convergence results for the expected sub-optimality of the final iterate of SGD for minimizing non-smooth convex functions in various settings. GD denotes the sub-gradient descent method, and lower bounds of GD also hold for SGD. The lower bounds for Lipschitz convex functions in Shamir & Zhang (2013); Harvey et al. (2019a) can also be extended to fixed step size $1/\sqrt{T}$, observed by Koren & Segal (2020).

and interpretable form of SGD. If there is a lower bound saying the last iterate of SGD is worse than the running average, we may need to compare the last iterate and running average when implementing the algorithm.

A line of works attempts to understand the convergence rate of the final iterate of SGD. A seminar work Shamir & Zhang (2013) first established a near-optimal $O(\log T/\sqrt{T})$ convergence rate for the final iterate of SGD with a STANDARD step size schedule $\eta_t = 1/\sqrt{t}$. Jain et al. (2019) proved an information-theoretically optimal $O(1/\sqrt{T})$ upper bound using a rather NON-STANDARD step size schedule. Roughly speaking, the $T$ steps are divided into $\log T$ phases, and the step size decreases by half when entering the next phase. Many implementations take ever-shrinking step sizes, which is somewhat consistent with this theoretical result. Harvey et al. (2019a) gave an $\Omega(\log T/\sqrt{T})$ lower bound for the STANDARD $\eta_t = 1/\sqrt{t}$ step size schedule, but their construction requires the dimension $d$ to be no less than $T$, which is restrictive. See Table 1 for more details. A natural question arises:

**Question:** *What's the dependence on dimension $d$ of the convergence rate of SGD's final iterate with standard step sizes when $d \leq T$?*

In a recent COLT open question raised by Koren & Segal (2020), the same problem was posed but mainly for the more restrictive constant dimension setting. Moreover, they conjectured that the right convergence rate of SGD with standard step size in the **constant** dimensional case is $\Theta(1/\sqrt{T})$. As preliminary support evidence for their conjecture, they analyzed a one-dimensional one-sided random walk special case. However, this result is limited in the one-dimension setting for the particular absolute-value function and thus can not be easily generalized. Analyzing the final-iterate convergence rate of SGD in the general dimension for general convex functions is a more exciting and challenging question. In particular, in Koren & Segal (2020), they wrote:

*For dimension $d > 1$, a natural conjecture is that the right convergence rate is $\Theta(\log d/\sqrt{T})$, but we have no indication to corroborate this.*

Motivated by this, we mainly focus on analyzing the final iterate of SGD with standard step size in general dimension $d \leq T$ without smoothness assumptions in this paper.

## 1.1 OUR CONTRIBUTIONS

Our first main result is an $\Omega(\log d/\sqrt{T})$ lower bound for SGD minimizing Lipschitz convex functions with a fixed step size $\eta_t = 1/\sqrt{T}$ when dimension $d \leq T$, generalizing the result in Harvey et al. (2019a). Our main observation is that we can let the initial point $x_1$ stay still for any number of steps as long as $\mathbf{0}$ is one of the sub-gradient of $f$ at $x_1$. By modifying the original construction of Harvey et al. (2019a), we can keep $x_1$ at $\mathbf{0}$ for $T - d$ steps and then 'kick' it to start taking a similar route as in Harvey et al. (2019a) in a $d$-dimensional space, which incurs an $\Omega(\log d/\sqrt{T})$ sub-optimality.

This result is generalized to Lipschitz convex functions with $1/\sqrt{t}$ decreasing step size schedule with the same sub-optimality, and an $\Omega(\log d/T)$ lower bound to strongly convex functions with $1/t$ step size schedule is also constructed with the similar technique. Our lower bound results imply that the last iterate with fixed step size has sub-optimal convergence rate for SGD in general theoretically, which, unfortunately, is used a lot in practice.

As for the upper bound, we present a new method based on martingale and Freedman's inequality to analyze the one-dimensional case. Though seemingly straightforward, the convergence rate of fixed-step-size SGD for one-dimensional linear functions is still open and non-trivial. Koren & Segal (2020) considered minimizing a linear function with a restricted SGD oracle which only outputs $\pm 1$, reducing this problem to a one-sided random walk. We relax the restriction on the SGD oracle and prove an $O(1/\sqrt{T})$ optimal rate for a class of convex functions which we call nearly linear convex functions, with the help of martingale theory. The class of nearly linear functions captures many common functions, such as linear functions, $|x|, e^x, x^2 + x, -\sin(x)$ on $[0, 1]$.

Our contributions are summarized as follows:

- We prove an $\Omega(\log d/\sqrt{T})$ lower bound for the sub-optimality of the final iterate of SGD minimizing non-smooth Lipschitz convex functions with $\eta_t = 1/\sqrt{T}$ step size schedule. We also generalize this bound to the $\eta_t = 1/\sqrt{t}$ decreasing step size schedule, and also prove an $\Omega(\log d/T)$ lower bound for non-smooth strongly convex functions with $\eta_t = 1/t$. To the best of our knowledge, our results are the first that characterize the general dimension dependence in analyzing the final iterate convergence of SGD with standard step sizes.
- We prove an optimal $O(1/\sqrt{T})$ upper bound for the sub-optimality of the final iterate of SGD minimizing nearly linear Lipschitz convex functions with fixed $\Theta(1/\sqrt{T})$ step sizes in one dimension, which captures a broad class of convex functions including linear functions.

## 2 PRELIMINARIES

Given a bounded convex set $\mathcal{K} \subset \mathbb{R}^d$, and a convex function $f : \mathcal{K} \to \mathbb{R}$ defined on $\mathcal{K}$, our goal is to solve $\min_{x \in \mathcal{K}} f(x)$. In the black-box optimization, there is no explicit representation of $f$. Instead, we can use a stochastic oracle to query the sub-gradients of $f$ at $x \in \mathcal{K}$. The set $\mathcal{K}$ is given in the form of a projection oracle, which outputs the closest point in $\mathcal{K}$ to a given point $x$ in the Euclidean norm. We introduce several standard definitions.

**Definition 1** (Sub-gradient). *A sub-gradient $g \in \mathbb{R}^d$ of a convex function $f : \mathcal{K} \to \mathbb{R}$ at point $x$, is a vector satisfying that for any $x' \in \mathcal{K}$,*

$$f(x') - f(x) \geq g^\top (x' - x). \tag{2}$$

*We use $\partial f(x)$ to denote the set of all sub-gradients of $f$ at $x$.*

**Definition 2** (Strong Convexity). *A function $f : \mathcal{K} \to \mathbb{R}$ is said to be $\alpha$-strongly convex, if for any $x, y \in \mathcal{K}$ and $g \in \partial f(x)$, the following holds:*

$$f(y) - f(x) \geq g^\top (y - x) + \frac{\alpha}{2}\|y - x\|_2^2 \tag{3}$$

**Definition 3** (Lipschitz Function). *A function $f : \mathcal{K} \to \mathbb{R}$ is called $G$-Lipschitz (with respect to $\ell_2$ norm), if for any $x, y \in \mathcal{K}$, we have that:*

$$|f(x) - f(y)| \leq G\|x - y\|_2 \tag{4}$$

*Further, if we assume $f$ is convex, the above definition is equal to $\|g\|_2 \leq G$ for any sub-gradient $g$.*

Let $\Pi_\mathcal{K}$ denote the projection operator on $\mathcal{K}$, the (projected) stochastic gradient descent (SGD) is described in Algorithm 1. We make the following standard assumption on the convex objective $f$ and the SGD algorithm we consider throughout this paper:

**Assumption 1** (Standard Assumption). *We make the following assumptions for the objective $f$ and running SGD:*

- *The domain $\mathcal{K} \subset \mathbb{R}$ is convex and bounded with diameter $D$.*

---

**Algorithm 1** Stochastic gradient descent with the final iterate output

---

1: Given $\mathcal{K} \subset \mathbb{R}^d$, initial point $x_1 \in \mathcal{K}$, step size schedule $\eta_t$:
2: **for** $j = 1, ..., T$: **do**
3:     Query stochastic gradient oracle at $x_t$ for $\hat{g}_t$ such that $\mathbb{E}[\hat{g}_t | \hat{g}_1, ..., \hat{g}_{t-1}] \in \partial f(x_t)$
4:     $y_{t+1} = x_t - \eta_t \hat{g}_t$
5:     $x_{t+1} = \Pi_{\mathcal{K}}(y_{t+1})$
6: **end for**
7: **return** $x_{T+1}$

---

- *The objective $f : \mathcal{K} \to \mathbb{R}$ is convex and G-Lipschitz, and not necessarily differentiable.*

- *The output stochastic gradients are bounded: $\|\hat{g}_t\|_2 \leq G$, and we have $\mathbb{E}[\hat{g}_t \mid \hat{g}_1, \cdots, \hat{g}_{t-1}] \in \partial f(x_t)$.*

The first two items hold for both our lower bound and upper bound. Our results are in the strong versions regarding the third item. In particular, our lower bound even holds for Gradient Descent (GD), i.e., even if the gradient oracle always outputs $\hat{g}_t \in \partial f(x_t)$ rather than in expectation, one still has the lower bound $\Omega(\log d/\sqrt{T})$. Our upper bound works for the SGD, where the oracle's outputs can be stochastic and one only assumes their expectations are sub-gradients.

## 3 LOWER BOUNDS

In this section we prove our main result, that is the final iterate of SGD for (non-smooth) Lipschitz convex functions with fixed step sizes $\eta_t = 1/\sqrt{T}$ has sub-optimality $\Omega(\log d/\sqrt{T})$, even with deterministic oracle. We build upon the construction in Harvey et al. (2019a), which is a variant of classical lower bound constructions Nesterov (2003) and proves an $\Omega(\log T/\sqrt{T})$ lower bound for the high-dimensional case $d \geq T$.

In a nutshell, we consider the setting $d \leq T$ and construct a function $f$ along with a special sub-gradient oracle such that the initial point stays still for the first $T - d$ steps, and then start moving in Algorithm 1, in which the final iterate satisfies $f(x_{T+1}) = \Omega(\log d/\sqrt{T})$. Then we extend the analysis to decreasing step sizes and strongly convex functions.

Let $[j]$ be the set of positive integers no larger than $j$. For simplicity, we consider convex functions over the $d$-dimensional Euclidean unit ball. Let $\mathbf{0}$ be the $d$-dimensional all-zero vector. We present our proof for general convex functions with fixed step sizes first. For decreasing step sizes and strongly convex functions, it is straightforward to scale our construction and get corresponding lower bounds, and we leave the proofs in the Appendix.

**Theorem 4.** *For any positive integer $T > 0$ and $1 \leq d \leq T$, there exists a 1-Lipschitz convex function $f : \mathcal{K} \to \mathbb{R}$ where $\mathcal{K} \subset \mathbb{R}^d$ is the Euclidean unit ball, and a non-stochastic sub-gradient oracle satisfying Assumption 1, such that when executing Algorithm 1 on $f$ with initial point $\mathbf{0}$ and step size schedule $\eta_t = 1/\sqrt{T}$, the last iterate satisfies:*

$$f(x_{T+1}) - \min_{x \in \mathcal{K}} f(x) \geq \frac{\log d}{32\sqrt{T}} \tag{5}$$

*Proof.* Let $\mathcal{B}_d$ be the Euclidean unit ball and define $f : \mathcal{B}_d \to \mathbb{R}$ for $i \in [d + 1] \cup \{0\}$ to be:

$$f(x) = \max_{0 \leq i \leq d+1} H_i(x)$$

where $H_i(x) = h_i^\top x$, and we define for $i \geq 1$

$$h_{i,j} = \begin{cases} a_j & (\text{ if } 1 \leq j < i) \\ -b_i & (\text{ if } i = j \leq d) \\ 0 & (\text{ if } i < j \leq d) \end{cases} \quad \text{and} \quad a_j = \frac{1}{8(d+1-j)}, \quad b_j = \frac{1}{2} \quad (\text{ for } j \in [d])$$

in which $h_{i,j}$ is the $j$-th coordinate of $h_i$. Additionally, let $h_0 = \mathbf{0}$ and $H_0(x) = 0$. It's straightforward to check that $f$ is 1-Lipschitz on $\mathcal{K}$, with a minimum value of 0. Furthermore, $\partial f(x)$ is the convex

hull of $\{h_i \mid i \in \mathcal{I}(x)\}$ where $\mathcal{I}(x) = \{i \geq 0 \mid H_i(x) = f(x)\}$, which is a standard fact in convex analysis Hiriart-Urruty & Lemaréchal (2013).

Setting $x_1 = \mathbf{0}$, we observe that $f(x_1) = 0$ which attains the global minimum, and by the characterization of $\partial f(x)$ from above, we know that $h_0 = \mathbf{0}$ is a sub-gradient at $x_1$. This observation allows our non-stochastic sub-gradient oracle to output $\mathbf{0}$ for the first $T - d$ steps and outputs $h_{i'}$ where $i' = \min \mathcal{I}(x) \setminus \{0\}$ for the last $d$ steps. Define $z_1 = \cdots = z_{T-d+1} = 0$, let $T^* =: T - d$ and we further define

$$z_{t,j} = \begin{cases} \frac{b_j}{\sqrt{T}} - a_j \frac{t-j-T^*-1}{\sqrt{T}} & (\text{ if } 1 \leq j < t - T^*) \\ 0 & (\text{ if } t - T^* \leq j \leq d) \end{cases} \quad (\text{ for } t > T^* + 1).$$

We show inductively that these are precisely the first $T$ iterates produced by algorithm 1 when using the sub-gradient oracle defined above. The following claim is easy to verify from the definition.

**Claim 5.** *We have the following claims:*

- $z_t$ *is non-negative. In particular,* $z_{t,j} \geq \frac{1}{4\sqrt{T}}$ *for* $j < t - T^*$ *and* $z_{t,j} = 0$ *for* $j \geq t - T^*$.

- $z_{t,j} \leq \frac{1}{2\sqrt{T}}$ *for all* $j$. *In particular,* $z_t \in \mathcal{K}$.

*Proof.* It is evident that $z_{t,j} = 0$ for $j \geq t - T^*$ from the definition. As $\frac{b_j}{\sqrt{T}} = \frac{1}{2\sqrt{T}}$, it suffices to prove that $0 \leq a_j \frac{t-j-T^*-1}{\sqrt{T}} \leq \frac{1}{4\sqrt{T}}$, which is direct as $0 \leq t - j - T^* - 1 \leq d + 1 - j$. $\square$

We can now determine the value and sub-gradient at $z_t$. The case for the first $T^*$ steps is trivial as the sub-gradient oracle always outputs $\mathbf{0}$ and $x_1$ never moves a bit. For the last $d$ steps we have that $z_t$ is supported on its first $t - T^*$ coordinates, and $h_{t-T^*}^\top z_t = h_{i-T^*}^\top z_t$ for all $i > t > T^*$.

For the other case $T^* + 1 \leq i < t$, one has that

$$z_t^\top (h_{t-T^*} - h_{i-T^*}) = \sum_{j=i-T^*}^{t-T^*} z_{t,j}(h_{t-T^*,j} - h_{i-T^*,j}) = \sum_{j=i-T^*}^{t-T^*-1} z_{t,j}(h_{t-T^*,j} - h_{i-T^*,j})$$

$$= \sum_{j=i-T^*+1}^{t-T^*-1} z_{t,j}(h_{t-T^*,j} - h_{i-T^*,j}) + z_{t,i-T^*}(h_{t-T^*,i-T^*} - h_{i-T^*,i-T^*})$$

$$= \sum_{j=i-T^*+1}^{t-T^*-1} z_{t,j} a_j + z_{t,i-T^*}(a_{i-T^*} + 1/2) > 0$$

which means $z_t^\top h_{t-T^*} > z_t^\top h_{i-T^*}$ for all $T^* + 1 \leq i < t$.

The two results together guarantee that $H_{t-T^*}(z_t) \geq H_{i-T^*}(z_t)$ for all $T^* + 1 \leq i$ and further $f(z_t) = H_{t-T^*}(z_t)$. Combining with the fact $\mathcal{I}(z_t) = \{t - T^*, ..., d + 1\}$, we conclude that the sub-gradient oracle outputs $h_{t-T^*}$ at time $t$.

**Lemma 6.** *For the function constructed in this section, the solution of $t$-th step in algorithm 1 equals to $z_t$ for every $T^* < t \leq T + 1$.*

*Proof.* We prove this lemma by induction. For base case $t = T^* + 1$, we know that $z_t = \mathbf{0} = x_t$ holds. Next, when $z_t = x_t$ holds for some $t$:

$$y_{t+1,j} = z_{t,j} - \frac{1}{\sqrt{T}} h_{t-T^*,j}$$

$$= \begin{cases} \frac{b_j}{\sqrt{T}} - a_j \frac{t-j-T^*-1}{\sqrt{T}} & (\text{ for } 1 \leq j < t - T^*) \\ 0 & (\text{ for } j \geq t - T^*) \end{cases} - \frac{1}{\sqrt{T}} \begin{cases} a_j & (\text{ if } 1 \leq j < t - T^*) \\ -b_i & (\text{ if } t - T^* = j \leq d) \\ 0 & (\text{ if } t - T^* < j \leq d) \end{cases}$$

$$= \begin{cases} \frac{b_j}{\sqrt{T}} - a_j \frac{t-j-T^*}{\sqrt{T}} & (\text{ for } j < t - T^*) \\ \frac{b_t}{\sqrt{T}} = \frac{1}{2\sqrt{T}} & (\text{ for } j = t - T^*) \\ 0 & (\text{ for } j > t - T^*) \end{cases}.$$

So $y_{t+1} = z_{t+1}$. Since $z_{t+1} \in \mathcal{K}$, we have that $x_{t+1} = z_{t+1}$. $\qquad\square$

From the above equivalence, we have that the vector $x_t$ in algorithm 1 is equal to $z_t$ for $t \in [T+1]$, which allows the determination of the value of the final iterate:

$$f(x_{T+1}) = f(z_{T+1}) = H_{d+1}(z_{T+1}) \geq \sum_{j=1}^{d} h_{d+1,j} z_{T+1,j} \geq \sum_{j=1}^{d} \frac{1}{8(d+1-j)} \frac{1}{4\sqrt{T}} > \frac{\log d}{32\sqrt{T}}.$$

$\qquad\square$

**Remark 7.** *For the case $d = 1$ we still have the $\Omega(1/\sqrt{T})$ lower bound, by not using $\sum_{i=1}^{d} \frac{1}{i} > \log d$ in the last step.*

**Remark 8.** *Notably, our lower bound is valid even for GD, where one can access a noiseless sub-gradient oracle.*

Theorem 4 improves the previously known lower bound by a factor of $\log d$, implying an inevitable dependence on the dimension of the convergence of SGD's final iterate. Though our proof is built upon Harvey et al. (2019a), their construction doesn't apply directly. Other natural ways of adaption, for example, cyclic (gradient oracle repeatedly goes over each coordinate), repeated (gradient oracle stays at one coordinate for $T/d$ steps then go to the next), do not work here.

Next, we extend this result to Lipschitz convex functions with step sizes $\eta_t = \frac{1}{\sqrt{t}}$ and strongly convex functions with step sizes $\eta_t = \frac{1}{t}$, both known to be the optimal choice of learning rate schedule. The proofs are mostly similar to that of Theorem 4, and we defer them to the Appendix.

**Corollary 9.** *For any $T$ and $1 \leq d \leq T$, there exist a 1-Lipschitz convex function $f : \mathcal{K} \to \mathbb{R}$ where $\mathcal{K} \subset \mathbb{R}^d$ is the Euclidean unit ball, and a non-stochastic sub-gradient oracle satisfying Assumption 1, such that when executing algorithm 1 on $f$ with initial point $\mathbf{0}$ and step size schedule $\eta_t = 1/\sqrt{t}$, the last iterate satisfies:*

$$f(x_{T+1}) - \min_{x \in \mathcal{K}} f(x) \geq \frac{\log d}{32\sqrt{T}} \tag{6}$$

**Corollary 10.** *For any $T$ and $1 \leq d \leq T$, there exist a 3-Lipschitz and 1-strongly convex function $f : \mathcal{K} \to \mathbb{R}$ where $\mathcal{K} \subset \mathbb{R}^d$ is the Euclidean unit ball, and a non-stochastic sub-gradient oracle satisfying Assumption 1, such that when executing Algorithm 1 on $f$ with initial point $\mathbf{0}$ (the global minimum) and step size schedule $\eta_t = 1/t$, the final iterate satisfies:*

$$f(x_{T+1}) - \min_{x \in \mathcal{K}} f(x) \geq \frac{\log d}{5T} \tag{7}$$

## 4 UPPER BOUND IN ONE DIMENSION

With our lower bound, it is natural to conjecture that the optimal rate should be $\Theta(\log d/\sqrt{T})$ when $d \leq T$. In particular, it's believed that in the one-dimensional case, the optimal rate is $\Theta(1/\sqrt{T})$.

As mentioned in the introduction, Koren & Segal (2020) considered a random walk induced by a linear function as evidence for this conjecture in one dimension, which is somewhat restricted. In this section, we relax their assumptions by considering a function class that we call nearly linear functions, which capture a broad class of functions, including linear functions, and prove an optimal rate $O(1/\sqrt{T})$. For the general Lipschitz convex function class, our analysis also recovers the previously known best bound $O(\log T/\sqrt{T})$.

### 4.1 NEARLY LINEAR FUNCTIONS

Let $f^* = \min_{x \in \mathcal{K}} f(x)$. We need the following definition before defining nearly linear functions.

**Definition 11.** *We say a point $x$ is good if $f(x) - f^* \leq \frac{4GD}{\sqrt{T}}$, and define a set of good points by $\mathcal{S}$:*

$$\mathcal{S} = \{x \in \mathcal{K} : f(x) - f^* \leq \frac{4GD}{\sqrt{T}}\}.$$

Now we can define the convex function family. In a nutshell, the class of nearly linear functions we consider is the function such that at any not-good point, the absolute value of its sub-gradient is not too small. Put it formally:

**Definition 12** (Nearly Linear Function). *We call a convex function $f : \mathcal{K} \to \mathbb{R}$ nearly linear if there exist a constant $0 < c \leq 1$, such that for any $x_t \notin \mathcal{S}$ which does not belong to the set of good points, we have $\left|\mathbb{E}[\hat{g}_t \mid \hat{g}_1, \cdots, \hat{g}_{t-1}]\right| \in [cG, G]$.*

We note that any general Lipschitz convex function is nearly linear with $c = 1/\sqrt{T}$, and our later analysis recovers the previously known best bound $O(\log T/\sqrt{T})$ under this interpretation. Therefore our method is a strict improvement over previous results.

The family of nearly linear functions captures those functions whose sub-gradients do not change drastically outside the set of good points, for example, $|x|, e^x, x^2 + x, -\sin(x)$. The linear functions considered in Koren & Segal (2020) lie in this family. The nice property of nearly linear functions allows a martingale-based analysis which gives an improved $O(1/\sqrt{T})$ bound.

Our proof is based on the Martingale (difference) (See Appendix for a detailed definition), and we use Freedman's Inequality given below.

**Theorem 13** (Freedman's Inequality, Theorem 1.6 in Freedman (1975)). *Consider a real-valued martingale difference sequence $\{X_t\}_{t \geq 0}$ such that $X_0 = 0$, and $\mathbb{E}[X_{t+1}|\mathcal{F}_t] = 0$ for all $t$, where $\{\mathcal{F}_t\}_{t \geq 0}$ is the filtration defined by the sequence. Assume that the sequence is uniformly bounded, i.e., $|X_t| \leq M$ almost surely for all $t$. Now define the predictable quadratic variation process of the martingale to be $W_t = \sum_{j=1}^{t} \mathbb{E}[X_j^2|\mathcal{F}_{j-1}]$ for all $t \geq 1$. Then for all $\ell \geq 0$ and $\sigma^2 > 0$ and any stopping time $\tau$, we have*

$$\Pr\left[\left|\sum_{j=0}^{\tau} X_j\right| \geq \ell \wedge W_\tau \leq \sigma^2 \text{for stopping time } \tau\right] \leq 2\exp\left(-\frac{\ell^2/2}{\sigma^2 + M\ell/3}\right).$$

Some previous works also use martingale theory to analyze SGD. For example, Harvey et al. (2019a) generalizes Freedman's Inequality to demonstrate a high probability (w.p. $1 - \delta$) suboptimality bound $O(\log(1/\delta)\log T/\sqrt{T})$ for SGD with standard step sizes, which is improved to $O(\log(1/\delta)/\sqrt{T})$ by Harvey et al. (2019b).

### 4.2 ANALYSIS

We show how to improve the convergence of the last iterate of SGD with a fixed step size $\eta = \frac{4D}{G\sqrt{T}}$ in one dimension for nearly linear functions. The proof mainly consists of two parts. In the first part, we prove that for running SGD with fixed step sizes for any convex function satisfying Assumption 1, with very high probability, the solution goes into the set of good points at least once. In some sense, this is consistent with the known result that averaging scheme can achieve the optimal rate. It is straightforward to get the following lemma by convexity.

**Lemma 14.** *For any $x \in \mathcal{K} \setminus \mathcal{S}, \forall \nabla f(x) \in \partial f(x)$, one has*

$$|\nabla f(x)| > \frac{G}{\sqrt{T}}.$$

Suppose we start from an arbitrary point $x_1 \in \mathcal{K}$ and the (random) sequence of the SGD algorithm with the fixed step size $\eta$ is denoted by $x_1, x_2, \cdots, x_{T+1}$, i.e. $x_{t+1} = \Pi_{\mathcal{K}}(x_t - \eta\hat{g}_t)$. The following lemma says that with a very high probability, the solution enters $\mathcal{S}$ at least once.

**Lemma 15.** *Given any $x_1 \in \mathcal{K}$, and let $\eta = \frac{4D}{G\sqrt{T}}$. For any nearly linear function $f$ under the Assumption 1, define $\tau_t := \infty$ if SGD never goes back to $\mathcal{S}$ in the first $t$ steps, and $\tau_t := \min_i\{1 \leq i \leq t \mid x_i \in \mathcal{S}\}$ otherwise. If $t \geq T + 1$ and $k \geq 10$, we have that*

$$\Pr[\tau_t = \infty \mid x_1] \leq 2\exp(-\Omega(T)).$$

Lemma 15 shows that the probability that $x_t$ never entered $\mathcal{S}$ in the first $T$ steps is negligible, whose proof can be found in the Appendix.

In the second part, we bound the tail probability of the sub-optimality of the last iterate for nearly linear functions, from which we can bound the expectation of the sub-optimality. Roughly speaking, we consider the events that $f(x_{T+1}) - f^* \geq \frac{GDk}{\sqrt{T}}$ and the last $T + 1 - i$ steps all lie out the set of good points, and bound its probability by $\exp\left(-\Omega(k + (T + 1 - i))\right)$. And by Union Bound we know that the tail probability $\Pr[f(x_{T+1}) - f^* \geq \frac{GDk}{\sqrt{T}}] \leq \exp(-\Omega(k))$, which is enough to get the optimal bound $O(\frac{GD}{\sqrt{T}})$.

**Theorem 16.** *Given positive integer $T > 0$ which is large enough, running SGD with a fixed step size $\eta = \frac{4D}{G\sqrt{T}}$ on any nearly linear function $f$ under Assumption 1 for $T$ steps, one has*

$$\mathbb{E}[f(x_{T+1}) - f^*] = O(\frac{GD}{\sqrt{T}}),$$

*where $f^* = \min_{x \in \mathcal{K}} f(x)$.*

*Proof.* We try to bound the tail probability, that is $\Pr[f(x_{T+1}) - f^* \geq \frac{GDk}{\sqrt{T}}]$ for any $k \geq 10$. We define $t := \infty$ if SGD never goes in the set $\mathcal{S}$ and let $t := \max_i \{1 \leq i \leq T + 1 \mid x_i \in \mathcal{S}\}$ otherwise. One has

$$\Pr[f(x_{T+1}) - f^* \geq \frac{GDk}{\sqrt{T}}]$$

$$= \sum_{i=1}^{T+1} \Pr[f(x_{T+1}) - f^* \geq \frac{GDk}{\sqrt{T}} \wedge t = i] + \Pr[f(x_{T+1}) - f^* \geq \frac{GDk}{\sqrt{T}} \wedge t = \infty]$$

$$= \sum_{i=1}^{T} \Pr[f(x_{T+1}) - f^* \geq \frac{GDk}{\sqrt{T}} \wedge t = i] + \Pr[f(x_{T+1}) - f^* \geq \frac{GDk}{\sqrt{T}} \wedge t = \infty],$$

where the second equality follows from the fact that $\Pr[f(x_{T+1}) - f^* \geq \frac{GDk}{\sqrt{T}} \wedge t = T + 1] = 0$ by the definition of $\mathcal{S}$ and $k \geq 10$. By Lemma 15, we have

$$\Pr[f(x_{T+1}) - f^* \geq \frac{GDk}{\sqrt{T}} \wedge t = \infty] \leq \Pr[t = \infty] \leq 2\exp(-\Omega(T)),$$

which is negligible when $T$ is large enough.

Now we begin to bound $\Pr[f(x_{T+1}) - f^* \geq \frac{GDk}{\sqrt{T}} \wedge t = i]$. We use $y_i = x_i - x_{i-1}$ to capture the movement of the solution. Let $n_L = \inf_{x \in \mathcal{S}} x$ and $n_R = \sup_{x \in \mathcal{S}} x$, which exist because the domain is bounded and the function is continuous. By Definition 11, there exists $x^* \in \arg\min_{x \in \mathcal{K}} f(x)$ such that either $|n_R - x^*| \geq 4D/\sqrt{T}$ or $|n_L - x^*| \geq 4D/\sqrt{T}$. By our setting of step size $\eta$, if $x_j > n_R$ for some $j$, it is impossible that $x_{j+1} < n_L$, and vice versa.

Consider the event $t = i$ and assume $x_{i+1} > n_R$ first. Hence $x_j > n_R$ for all $i < j \leq T + 1$. By the Assumption that $f$ is nearly linear, we have $\mathbb{E}[y_i] \in [-\eta G, -c\eta G]$ for some constant $c \in (0, 1]$ (See Definition 11). Let $\mathcal{F}_{i-1}$ be the filtration and $\tilde{y}_i = y_i - \mathbb{E}[y_i \mid \mathcal{F}_{i-1}]$. Obviously, we know $\mathbb{E}[\tilde{y}_i \mid \mathcal{F}_{i-1}] = 0$, $|\tilde{y}_i| \leq 2\eta G$ and $\{\tilde{y}_i\}$ is a martingale difference sequence. We know that $W_{(i,T+1]} := \sum_{j=i+1}^{T+1} \mathbb{E}[\tilde{y}_i^2 \mid \mathcal{F}_{i-1}] \leq \sum_{j=i+1}^{T+1} \mathbb{E}[y_i^2 \mid \mathcal{F}_{i-1}] \leq \eta^2 G^2 (T + 1 - i)$ as $|y_i| \leq \eta G$. Let $\ell = \sum_{j=i+1}^{T+1} \mathbb{E}[y_j \mid \mathcal{F}_{j-1}]$. It is evident that $\ell \leq -c\eta G(T + 1 - i)$ by the assumption of being nearly linear and $t_j > n_R$.

Condition on $f(x_{T+1}) - f^* \geq \frac{GDk}{\sqrt{T}} \wedge t = i$. It follows that $\sum_{j=i+1}^{T+1} y_i \geq \frac{D(k-4)}{\sqrt{T}}$. More specifically, as $x_i \in \mathcal{S}$ and thus $f(x_i) - f^* \leq \frac{4GD}{\sqrt{T}}$, we have that $f(x_{T+1}) - f(x_i) \geq \frac{GD(k-4)}{\sqrt{T}}$ and further $x_{T+1} - x_i = \sum_{j=i+1}^{T+1} y_j \geq \frac{D(k-4)}{\sqrt{T}}$. Moreover, we know $W_{(i,T+1]} \leq \eta^2 G^2 (T + 1 - i)$ and $\ell \leq -c\eta G(T + 1 - i)$, which means $\sum_{j=i+1}^{T+1} \tilde{y}_j = \sum_{j=i+1}^{T+1} y_j - \sum_{j=i+1}^{T+1} \mathbb{E}[y_j \mid \mathcal{F}_{j-1}] \geq \frac{D(k-4)}{\sqrt{T}} - \ell = \frac{D(k-4)}{\sqrt{T}} + |\ell|$.

As for the case when $x_{i+1} < n_L$, conditioning on $f(x_{T+1}) - f^* \geq \frac{GDk}{\sqrt{T}} \wedge t = i \wedge x_{i+1} < n_L$, it is similar to get $\sum_{j=i+1}^{T+1} \tilde{y}_j \leq -\frac{D(k-4)}{\sqrt{T}} - |\ell| \wedge W_{(i:T+1]} \leq \eta^2 G^2 (T + 1 - i) \wedge \ell \geq c\eta G(T + 1 - i)$

as well. Hence we have

$$\Pr[f(x_{T+1}) - f^* \geq \frac{GDk}{\sqrt{T}} \wedge t = i] \leq \Pr[|\sum_{j=i+1}^{T+1} y_j| \geq \frac{D(k-4)}{\sqrt{T}} \wedge t = i]$$

$$\leq \Pr[|\sum_{j=i+1}^{T+1} \tilde{y}_j| \geq \frac{D(k-4)}{\sqrt{T}} + |\ell| \wedge W_{(i:T+1]} \leq \eta^2 G^2(T+1-i) \wedge |\ell| \geq c\eta G(T+1-i)],$$

(8)

where the second inequality follows from the analysis above. Applying Freedman's Inequality (Theorem 13) over Equation (8), one has

$$\Pr[f(x_{T+1}) - f^* \geq \frac{GDk}{\sqrt{T}} \wedge t = i]$$

$$\leq \max_{|\ell| \geq c\eta G(T+1-i)} 2\exp\left(-\frac{(\frac{D(k-4)}{\sqrt{T}} + |\ell|)^2/2}{\eta^2 G^2(T+1-i) + 2\eta G(\frac{D(k-4)}{\sqrt{T}} + |\ell|)/3}\right)$$

$$\leq \max_{|\ell| \geq c\eta G(T+1-i)} 2\exp\left(-\frac{(\frac{D(k-4)}{\sqrt{T}} + |\ell|)/2}{\frac{\eta G}{c} + 2\eta G/3}\right)$$

$$\leq 2\exp\left(-\frac{3c}{10\eta G}(\frac{D(k-4)}{\sqrt{T}} + c\eta G(T+1-i))\right)$$

$$= 2\exp(-\frac{3c(k-4)}{40} - \frac{3}{10}c^2(T+1-i)).$$

Further, for $k \geq 10$, we have

$$\Pr[f(x_{T+1}) - f^* \geq \frac{GDk}{\sqrt{T}}]$$

$$= \sum_{i=1}^{T} \Pr[f(x_{T+1}) - f^* \geq \frac{GDk}{\sqrt{T}} \wedge t = i] + \Pr[f(x_{T+1}) - f^* \geq \frac{GDk}{\sqrt{T}} \wedge t = \infty]$$

$$\leq \sum_{i=1}^{T} 2\exp(-\frac{3c(k-4)}{40} - \frac{3}{10}c^2(T+1-i)) + 2\exp(-\Omega(\sqrt{T}))$$

$$\leq \frac{20}{3c^2}\exp(-\frac{3c(k-4)}{40}) + 2\exp(-\Omega(\sqrt{T})),$$

where the last step follows from the fact that for any constant $C > 0$ one has $\sum_{i=1}^{T}\exp(-Ci) \leq \int_{i=0}^{T-1}\exp(-Ci)\mathrm{d}i \leq 1/C$. As a result, for $h \geq 10GD/\sqrt{T}$, we have that

$$\Pr[f(x_{T+1}) - f^* \geq h] = O(\exp(-h\lambda)),$$

(9)

where $\lambda = \Theta(\frac{\sqrt{T}}{GD})$. Our conclusion follows from

$$\mathbb{E}[f(x_{T+1}) - f^*] = \int_0^{GD} \Pr[f(x_{T+1}) - f^* \geq h]\mathrm{d}h = O(1/\lambda) = O(\frac{GD}{\sqrt{T}}).$$

(10)

$\square$

## 5 CONCLUSION

In this paper, we analyze the final iterate convergence rate of SGD with standard step size schedules, proving $\Omega(\log d/\sqrt{T})$ and $\Omega(\log d/T)$ lower bounds for the sub-optimality of SGD minimizing non-smooth general convex and strongly convex functions respectively. We also prove a tight $O(1/\sqrt{T})$ upper bound for one-dimensional nearly linear functions, a more general setting than Koren & Segal (2020). This work is the first, to the best of our knowledge, that characterizes the dependence on dimension in the general $d \leq T$ setting, and we hope it can advance our theoretical understanding of the final iterate convergence of SGD with standard step sizes, and guide the implementations in practice.

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

## A MORE PRELIMINARIES

### A.1 PRELIMINARIES ON MARTINGALE

We demonstrate some basic definitions with a relationship to Martingale, which is used in the proof.

**Definition 17** (Martingale). *A sequence $Y_1, Y_2, \cdots$ is said to be a martingale with respect to another sequence $X_1, X_2, \cdots$ if for all $n$:*

- $\mathbf{E}\left(|Y_n|\right) < \infty$

- $\mathbf{E}\left(Y_{n+1} \mid X_1, \ldots, X_n\right) = Y_n$.

**Definition 18** (Martingale Difference). *Consider an adapted sequence $\{X_t, \mathcal{F}_t\}_{-\infty}^{\infty}$ on a probability space. $X_t$ is a martingale difference sequence (MDS) if it satisfies the following two conditions for all $t$:*

- $\mathbb{E}|X_t| < \infty$

- $\mathbb{E}[X_t \mid \mathcal{F}_{t-1}] = 0$, a.s.

**Definition 19** (Stopping Time). *A stopping time with respect to a sequence of random variables $X_1, X_2, X_3, \cdots$ is a random variable $\tau$ with the property that for each $t$, the occurrence or non-occurrence of the event $\tau = t$ depends only on the values of $X_1, X_2, X_3, \cdots, X_t$.*

## B OMITTED PROOFS FOR SECTION 3

### B.1 PROOF OF COROLLARY 9

*Proof.* Define $f : \mathcal{K} = \mathcal{B}_d \to \mathbb{R}$ and $h_i \in \mathbb{R}^d$ for $i \in [d+1] \cup \{0\}$ by

$$f(x) = \max_{0 \le i \le d+1} H_i(x)$$

where $H_i(x) = h_i^\top x$. For $i \ge 1$ we define

$$h_{i,j} = \begin{cases} a_j & (\text{ if } 1 \le j < i) \\ -b_i & (\text{ if } i = j \le d) \\ 0 & (\text{ if } i < j \le d) \end{cases} \quad \text{and} \quad a_j = \frac{1}{8(d+1-j)}, \quad b_j = \frac{\sqrt{j+T-d}}{2\sqrt{T}} \quad (\text{ for } j \in [d])$$

Additionally, let $h_0 = \mathbf{0}$ and $H_0(x) = 0$. It's easy to check that $f$ is 1-Lipschitz, with minimal value 0. We have that $\partial f(x)$ is the convex hull of $\{h_i \mid i \in \mathcal{I}(x)\}$ where $\mathcal{I}(x) = \{i \ge 0 \mid H_i(x) = f(x)\}$.

Our non-stochastic sub-gradient oracle outputs $\mathbf{0}$ for the first $T - d$ steps and outputs $h_{i'}$ where $i' = \min \mathcal{I}(x) \setminus \{0\}$ for the last $d$ steps. Define $z_1 = \cdots = z_{T-d+1} = \mathbf{0}$, let $T^* =: T - d$.

$$z_{t,j} = \begin{cases} \frac{b_j}{\sqrt{j+T^*}} - a_j \sum_{k=j+T^*+1}^{t-1} \frac{1}{\sqrt{k}} & (\text{ if } 1 \le j < t - T^*) \\ 0 & (\text{ if } t - T^* \le j \le d) \end{cases} \quad (\text{ for } t > T^* + 1).$$

We will show inductively that these are precisely the first $T$ iterates produced by algorithm 1 when using the sub-gradient oracle defined above. The following claim follows from definition.

**Claim 20.** *We have the following claims:*

- *$z_t$ is non-negative. In particular, $z_{t,j} \ge \frac{1}{4\sqrt{T}}$ for $j < t - T^*$ and $z_{t,j} = 0$ for $j \ge t - T^*$.*

- *$z_{t,j} \le \frac{1}{2\sqrt{T}}$ ofr all $j$. In particular, $z_t \in \mathcal{K}$.*

*Proof.* It is obvious that $z_{t,j} = 0$ for $j \ge t - T^*$ from the definition. As $\frac{b_j}{\sqrt{j+T^*}} = \frac{1}{2\sqrt{T}}$, it suffices to prove that $0 \le a_j \sum_{k=j+T^*}^{t-1} \frac{1}{\sqrt{k}} \le \frac{1}{4\sqrt{T}}$. We have that

$$0 \le \sum_{k=j+T^*}^{t-1} \frac{1}{\sqrt{k}} \le \int_{j+T^*-1}^{t-1} \frac{1}{\sqrt{x}} \mathrm{d}x = \frac{2(t-j-T^*)}{\sqrt{t-1} + \sqrt{j+T^*-1}} \le \frac{2(t-j-T^*)}{\sqrt{t-1}} \tag{11}$$

and further $\frac{t-j-T^*}{\sqrt{t-1}} \leq \frac{T+1-j-T^*}{\sqrt{T}} = \frac{d+1-j}{\sqrt{T}}$ by monotony. Thus $0 \leq a_j \sum_{k=j+T^*}^{t-1} \frac{1}{\sqrt{k}} \leq \frac{1}{4\sqrt{T}}$ follows from the definition of $a_j$.

$\square$

We can now determine the value and sub-differential at $z_t$. The case for the first $T^*$ steps is trivial as the sub-gradient oracle always outputs 0 and $x_1$ never moves a bit. For the last $d$ steps we have that $z_t$ is supported on its first $t - T^*$ coordinates and $h_{t-T^*}^\top z_t = h_{i-T^*}^\top z_t$ for all $i > t > T^*$.

For $T^* + 1 \leq i < t$, one has

$$z_t^\top(h_{t-T^*} - h_{i-T^*}) = \sum_{j=i-T^*}^{t-T^*} z_{t,j}(h_{t-T^*,j} - h_{i-T^*,j}) = z_{t,i}(a_i + 1) + \sum_{j=i+1}^{t-1} z_{t,j} a_j > 0.$$

which means $z_t^\top h_{t-T^*} > z_t^\top h_{i-T^*}$ for all $T^* + 1 \leq i < t$. The two results together guarantee that $H_{t-T^*}(z_t) \geq H_{i-T^*}(z_t)$ for all $T^* + 1 \leq i$ and further $f(z_t) = H_{t-T^*}(z_t)$. Combining with the fact $\mathcal{I}(z_t) = \{t - T^*, ..., d + 1\}$, we conclude that the sub-gradient oracle outputs $h_{t-T^*}$.

**Lemma 21.** *For the function constructed in this section, the solution of $t$-th step in algorithm 1 equals to $z_t$ for every $T^* < t \leq T + 1$.*

*Proof.* We prove this lemma by induction. For base case $t = T^* + 1$, we know that $z_t = \mathbf{0} = x_t$ holds. Next, when $z_t = x_t$ holds for some $t$:

$$y_{t+1,j} = z_{t,j} - \frac{1}{\sqrt{t}} h_{t-T^*,j}$$

$$= \left\{ \begin{array}{ll} \frac{b_j}{\sqrt{j+T^*}} - a_j \sum_{k=j+T^*}^{t-1} \frac{1}{\sqrt{k}} & (\text{ for } 1 \leq j < t - T^*) \\ 0 & (\text{ for } j \geq t - T^*) \end{array} \right\} - \frac{1}{\sqrt{t}} \left\{ \begin{array}{ll} a_j & (\text{ if } 1 \leq j < t - T^*) \\ -b_i & (\text{ if } t - T^* = j \leq d) \\ 0 & (\text{ if } t - T^* < j \leq d) \end{array} \right\}$$

$$= \left\{ \begin{array}{ll} \frac{b_j}{\sqrt{j+T^*}} - a_j \sum_{k=j+T^*}^{t} \frac{1}{\sqrt{k}} & (\text{ for } j < t - T^*) \\ \frac{b_t}{\sqrt{t}} = \frac{b_t}{\sqrt{j+T^*}} & (\text{ for } j = t - T^*) \\ 0 & (\text{ fro } j > t - T^*) \end{array} \right\}.$$

So $y_{t+1} = z_{t+1}$. Since $z_{t+1} \in \mathcal{K}$, we have that $x_{t+1} = z_{t+1}$.

$\square$

From the above claim we have that the vector $x_t$ in algorithm 1 is equal to $z_t$ for $t \in [T + 1]$, which allows determination of the value of the final iterate:

$$f(x_{T+1}) = f(z_{T+1}) = H_{d+1}(z_{T+1}) \geq \sum_{j=1}^{d} h_{d+1,j} z_{T+1,j} \geq \sum_{j=1}^{d} \frac{1}{8(d+1-j)} \frac{1}{4\sqrt{T}} > \frac{\log d}{32\sqrt{T}}.$$

$\square$

## B.2 PROOF OF COROLLARY 10

*Proof.* Define $f : \mathcal{K} = \mathcal{B}_d \to \mathbb{R}$ by $H_i \in \mathbb{R}^d$ for $i \in [d+1] \cup \{0\}$ to be:

$$f(x) = \max_{0 \leq i \leq d+1} H_i(x)$$

where $H_i(x) = h_i^\top x + \frac{1}{2}\|x\|^2$, and we define for $i \geq 1$

$$h_{i,j} = \left\{ \begin{array}{ll} a_j & (\text{ if } 1 \leq j < i) \\ -1 & (\text{ if } i = j \leq d) \\ 0 & (\text{ if } i < j \leq d) \end{array} \right. \quad \text{and} \quad a_j = \frac{1}{2(d+1-j)} \quad (\text{ for } j \in [d])$$

in which $h_{i,j}$ is the $j$-th coordinate of $h_i$. Additionally, let $h_0 = \mathbf{0}$ and $H_0(x) = \frac{1}{2}\|x\|^2$. It's straightforward to check that $f$ is 3-lipschitz and 1-strongly convex on $\mathcal{K}$, with minimal value 0. Furthermore, $\partial f(x)$ is the convex hull of $\{h_i + x \mid i \in \mathcal{I}(x)\}$ where $\mathcal{I}(x) = \{i \geq 0 \mid H_i(x) = f(x)\}$, a standard fact in convex analysis Hiriart-Urruty & Lemaréchal (2013).

Setting $x_1 = \mathbf{0}$, we observation that $f(x_1) = \mathbf{0}$ which attains the global minimum, and by the characterization of $\partial f(x)$ from above, we know that $h_0 + x_1 = \mathbf{0}$ is a sub-gradient at $x_1$. This observation allows our non-stochastic sub-gradient oracle to output $\mathbf{0}$ for the first $T - d$ steps and outputs $h_{i'} + x$ where $i' = \min \mathcal{I}(x) \setminus \{0\}$ for the last $d$ steps, since outputting $\mathbf{0}$ in the first $T - d$ steps makes $x_1 = ... = x_{T-d+1} = \mathbf{0}$ by the update rule of SGD. Define $z_1 = \cdots = z_{T-d+1} = \mathbf{0}$, let $T^* := T - d$ and

$$
z_{t,j} = \begin{cases} \frac{1-(t-T^*-j-1)a_j}{t-1} & (\text{ if } 1 \le j < t - T^*) \\ 0 & (\text{ if } t - T^* \le j \le T) \end{cases} \quad (\text{ for } t > T^* + 1).
$$

We will show inductively that these are precisely the first $T$ iterates produced by algorithm 1 when using the sub-gradient oracle defined above. The following claim is easy to verify from definition.

**Claim 22.** *We have the following claims:*

- $z_t$ *is non-negative. In particular,* $z_{t,j} \ge \frac{1}{2(t-1)}$ *for* $j < t - T^*$ *and* $z_{t,j} = 0$ *for* $j \ge t - T^*$.

- $z_t = \mathbf{0}$ *for* $t \in [T^* + 1]$ *and* $\|z_t\|^2 \le \frac{1}{t-1}$ *for* $t > T^* + 1$*. Thus* $z_t \in \mathcal{K}$ *for all* $t$.

*Proof.* The first claim simply follows from the fact that $\frac{t-T^*-j-1}{d-j+1} \le 1$. The second claim follows from that $(t - T^* - 1)\frac{1}{(t-1)^2} \le \frac{1}{t-1}$. $\square$

We can now determine the value and sub-differential at $z_t$. The case for the first $T^*$ steps is trivial as the sub-gradient oracle always outputs $\mathbf{0}$ and $x_1$ never moves. For the value of last $d$ steps we observe that $z_t$ is supported on its first $t - T^*$ coordinates by definition, and as a result $h_{t-T^*}^\top z_t = h_{i-T^*}^\top z_t$ for all $i > t > T^*$.

For the other case $T^* + 1 \le i < t$, one have that

$$
z_t^\top (h_{t-T^*} - h_{i-T^*}) = \sum_{j=i}^{t-1} z_{t,j}(h_{t-T^*,j} - h_{i-T^*,j}) = z_{t,i}(a_i + 1) + \sum_{j=i+1}^{t-1} z_{t,j} a_j > 0.
$$

which means $z_t^\top h_{t-T^*} > z_t^\top h_{i-T^*}$ for all $T^* + 1 \le i < t$. The two results together guarantee that $H_{t-T^*}(z_t) \ge H_{i-T^*}(z_t)$ for all $T^* + 1 \le i$ and thus $f(z_t) = H_{t-T^*}(z_t)$. Combining with the fact $\mathcal{I}(z_t) = \{t - T^*, ..., d + 1\}$, we conclude that the sub-gradient oracle outputs $h_{t-T^*} + z_t$.

**Lemma 23.** *For the function $f$ and its gradient oracle constructed in the proof, the output $x_t$ of $t$-th step in Algorithm 1 equals to $z_t$ for every $T^* < t \le T + 1$.*

*Proof.* We prove this lemma by induction. For base case $t = T^* + 1$, we know by the definition of $z_t$ that $z_t = 0 = x_t$ holds. Next, when $z_t = x_t$ for some $t$ holds, we have that

$$
y_{t+1,j} = z_{t,j} - \frac{1}{t}(h_{t-T^*,j} + z_{t,j})
$$

$$
= \frac{t-1}{t} \begin{cases} \frac{1-(t-T^*-j-1)a_j}{t-1} & (\text{ for } 1 \le j < t - T^*) \\ 0 & (\text{ for } j \ge t - T^*) \end{cases} - \frac{1}{t} \begin{cases} a_j & (\text{ if } 1 \le j < t - T^*) \\ -1 & (\text{ if } t - T^* = j \le d) \\ 0 & (\text{ if } t - T^* < j \le d) \end{cases}
$$

$$
= \frac{1}{t} \begin{cases} 1 - (t - T^* - j - 1)a_j & (\text{ for } 1 \le j < t - T^*) \\ 0 & (\text{ for } j \ge t - T^*) \end{cases} - \frac{1}{t} \begin{cases} a_j & (\text{ if } 1 \le j < t - T^*) \\ -1 & (\text{ if } t - T^* = j \le d) \\ 0 & (\text{ if } t - T^* < j \le d) \end{cases}
$$

$$
= \begin{cases} \frac{1-(t-T^*-j)a_j}{t} & (\text{ for } j < t - T^*) \\ \frac{1}{t} & (\text{ for } j = t - T^*) \\ 0 & (\text{ fro } j > t - T^*) \end{cases}.
$$

So $y_{t+1} = z_{t+1}$. Since $z_{t+1} \in \mathcal{K}$, we have that $x_{t+1} = z_{t+1}$.

$\square$

From the above equivalence we have that the vector $x_t$ in algorithm 1 is equal to $z_t$ for $t \in [T+1]$, which allows the determination of the value of the final iterate:

$$f(x_{T+1}) = f(z_{T+1}) = H_{d+1}(z_{T+1}) \geq \sum_{j=1}^{d} h_{d+1,j} z_{T+1,j} \geq \sum_{j=1}^{d} \frac{1}{2(d+1-j)} \frac{1}{2T} > \frac{\log d}{5T}.$$

□

## C    OMITTED PROOFS OF SECTION 4

### C.1    PROOF OF LEMMA 14

*Proof.* We prove this statement by contradiction. Suppose there exists $x \in \mathcal{K} \setminus \mathcal{S}$ such that $|\nabla f(x)| \leq \frac{G}{\sqrt{T}}$. By the convexity of $f$ and the definition of sub-gradient and let $x^* \in \mathcal{K}$ be a minimizer (arbitrarily if the minimizers are not unique), one has

$$f(x^*) \geq f(x) + \nabla f(x)(x - x^*),$$

which implies that

$$f(x) - f(x^*) \leq \nabla f(x)(x^* - x) \leq \frac{GD}{\sqrt{T}}.$$

This means $x \in \mathcal{S}$ and thus is a contradiction. □

### C.2    PROOF OF LEMMA 15

*Proof.* Let $n_L = \inf_{x \in \mathcal{S}} x$ and $n_R = \sup_{x \in \mathcal{S}} x$, which exist because the domain is bounded. By our setting of parameters and definition, we know if $x_j > n_R$, then it is impossible that $x_{j+1} < n_L$, and vice versa. As we are considering $\tau_t = \infty$, either $x_i > n_R$ for all $1 \leq i \leq t$, or $x_i < n_L$ for all $1 \leq i \leq t$. Without loss of generality, we consider first the case where $x_i > n_R$ for all $1 \leq i \leq t$. We define a random variable $y_i = x_i - x_{i-1}$ to capture the movement of the solution for $1 \leq i \leq t$.

Conditioning on $\tau = \infty$, i.e. $x_i > n_R$ for all $1 \leq i \leq t$, we have that $\mathbb{E}[y_i] \leq -c\eta G = -4D/T$ for $i \geq 2$ by Lemma 14 (the projection only makes the expectation smaller). By standard arguments, let $\mathcal{F}_i$ be the filtration and $\tilde{y}_i = y_i - \mathbb{E}[y_i \mid \mathcal{F}_{i-1}]$. It is easy to verify that $\{\tilde{y}_i\}$ is a martingale difference sequence:

$$\mathbb{E}[\tilde{y}_i \mid \mathcal{F}_i] = \mathbb{E}[y_i \mid \mathcal{F}_i] - \mathbb{E}[y_i \mid \mathcal{F}_i] = 0. \tag{12}$$
$$\mathbb{E}[|\tilde{y}_i|] \leq G\eta < \infty. \tag{13}$$

Obviously, one has $|\tilde{y}_i| \leq G\eta = \frac{4D}{\sqrt{T}}$ by the third line of Assumptions 1. As a result, $\mathbb{E}[\tilde{y}_i^2 \mid \mathcal{F}_{i-1}] = \mathbb{E}[y_i^2 \mid \mathcal{F}_{i-1}] - (\mathbb{E}[y_i \mid \mathcal{F}_{i-1}])^2 \leq \mathbb{E}[y_i^2 \mid \mathcal{F}_{i-1}] \leq \eta^2 G^2$. Hence, we get the estimation $W_t = \sum_{i=2}^{t} \mathbb{E}[\tilde{y}_i^2 \mid \mathcal{F}_{i-1}] \leq (t-1)\eta^2 G^2$. Let $\ell := \sum_{i=2}^{t} \mathbb{E}[y_i \mid \mathcal{F}_{i-1}]$. We know that $\ell = \sum_{i=1}^{t} \mathbb{E}[y_i \mid \mathcal{F}_{i-1}] \leq -t\eta cG$.

So far, we have shown that if $x_i > n_R$ for all $1 \leq i \leq t$, then we must have $\ell \leq -t\eta cG$, and at the same time $D \geq \sum_{i=2}^{t} y_i \geq -D$ which must happen. Similarly, if $x_i < n_L$ for all $1 \leq i \leq t$, then we have $\ell \geq t\eta cG$ and $-D \leq \sum_{i=2}^{t} y_i \leq D$. If we can show the probability that $|\ell| \geq t\eta cG$ and $|\sum_{i=2}^{t} y_i| \leq D$ happen simultaneously is small, we are done.

By the Freedman's Inequality, if $t = T + 1$, one has:

$$\Pr[\tau_t = \infty \mid x_1] \leq \Pr[x_i > n_R, 1 \leq i \leq t] + \Pr[x_i < n_L, 1 \leq i \leq t]$$

$$\leq 2 \Pr[|\sum_{i=2}^{t} \tilde{y}_i| \geq |\ell| - D \wedge |\ell| \geq t\eta cG]$$

$$= 2 \Pr[|\sum_{i=2}^{t} \tilde{y}_i| \geq |\ell| - D \wedge W_t \leq (t-1)\eta^2 G^2 \wedge |\ell| \geq t\eta cG]$$

$$\leq \max_{|\ell| \geq Tc\eta G} 2 \exp\left(-\frac{(\ell - D)^2}{T\eta^2 G^2 + \frac{4G\eta}{3}(\ell - D)}\right)$$

$$\leq 2 \exp(-\Omega(c^2 T)).$$

We complete the proof. $\qquad\qquad\qquad\qquad\qquad\qquad\qquad\qquad\qquad\qquad\qquad\qquad\qquad\qquad$ $\square$

