# OpenReview forum: "The Convergence Rate of SGD's Final Iterate: Analysis on Dimension Dependence"
_ICLR.cc/2023/Conference — Submitted to ICLR 2023_

### Official Review · Reviewer_6ixh · 2022-10-26

**Confidence:** 4
**Correctness:** 2
**Technical Novelty And Significance:** 3
**Empirical Novelty And Significance:** Not applicable
**Recommendation:** 5

**Clarity, Quality, Novelty And Reproducibility:**

In general, the paper is well-structured and easy to follows. However, some parts of the proofs require additional polishing and clarifications. The results are novel.

**Strength And Weaknesses:**

## Strengths

1. **Lower bound for the final iterate and fixed dimension.** This result is new for the literature and highlights the differences between the last-iterate and averaged-iterate convergence guarantees. It does not directly follow from the existing approaches for constructing lower bounds.

2. **Several classical stepsize schedules are considered.** The considered stepsize policies are among the most popular ones. This fact additionally justifies the importance of the derived results.


## Weaknesses

1. **Proofs contain a lot inaccuracies and missing parts.** I cannot verify the proof of Theorem 15, because it contains a number of inaccuracies and unexplained derivations. In the list of detailed comments, I provide all inaccuracies that I noticed. The authors should make the proofs much clearer according to my comments.

2. **Bounded domain.** The bounded domain assumption is quite restrictive. I have not found the place in the proof of the lower bounds where the authors rely on this (except for the strongly case, since without this assumption the class of the considered functions is empty). So, at least for Theorem 4 and Corollary 8 the authors can remove this assumption, if I am not missing anything. However, in the proof of the upper bound the authors do rely on the boundedness of the domain. This is a significant limitation.

3. **Bounded stochastic subgradients.** The third part of Assumption 1 is very restrictive, since it means that the noise is bounded. Is it possible to remove this assumption to get a similar upper bound?

## Detailed questions and comments

1. I think the paper will benefit from more in-depth comparison with the results from Jain et al. (2019). Although they have non-standard stepsizes, it will be good to provide them in this paper and discuss the differences with the standard ones.

2. The construction used in the proof of the lower bound significantly relies on the fact that the oracle acts in an adversarial manner. This is possible because of the non-smoothness of the problem. However, in many real problems the set of points where the objective is non-smooth has zeroth Lebesgue measure and the methods rarely occur in these points (especially if we consider stochastic methods). Moreover, the choice of the subgradient can be non-adversarial (e.g., one can try to choose the subgradient somehow randomly to avoid bad behavior). That is, it would be interesting to derive some lower bounds in less adversarial setups than in the one considered in this paper.

3. Page 5, the derivation after "For the other case $T^\ast + 1 \leq i < t$": I do not understand the second equality. Could the authors provided a detailed derivation? If I am not mistake, $h_{i - T^\ast, j} = 0$ for all $j > i - T^\ast$ and $h_{i - T^\ast, i - T^\ast} = - \frac{1}{2}$. I do not see how it is reflected in the derivation.

4. The discussion after Remark 7: one should emphasize that this lower bound is valid even for GD.

5. Proof of Theorem 15, the derivation after "One has": the first equality is not correct since the events are not independent. One should replace "$=$" with "$\leq$".

6. Proof of Theorem 15, "Without loss of generality": why the generality is not lost? I think it is possible to have $x_j > n_R$ and $x_{j+1} < n_L$ (due to the stochasticity). How the proof will work in this case?

7. Proof of Theorem 15, "for some constant $c = \Theta(1)$": this should be specified formally in the theorem.

8. Proof of Theorem 15, the upper bound for $W_{(i, T+1]}$: I do not understand how it is derived. Could the authors provided the detailed derivation? In particular, it is unclear why the absolute value is inside $|\cdot|$.

9. Proof of Theorem 15, the upper bound for $\ell$: I do not see how this upper bound is obtained. The authors should provide a detailed derivation.

10. The last step in formula (8) is unclear. The authors should provide a detailed derivation.

11. Page 9, "Applying Freedman's Inequality (Theorem 12)": Theorem 12 does not provide such inequality. The authors should provide the details on how Theorem 12 is applied here. Because of these inaccuracies, I did not check the next part going after "And further". The authors should proofread it as well.

12. Formula (10): why the integration is taken from $0$ to $GD$? Why this is $O(\lambda)$? The authors should provide a detailed derivation.

13. Appendix C.1: there is a mistake in the definition of sub-gradient.

14. Proof of Lemma 14, "Without loss of generality": similar question to 6.

15. Proof of Lemma 14, the derivation after "As a result": in then end one should have $\mathbb{E}[|y_i| \mid \mathcal{F}_i]$.

16. Proof of Lemma 14, lower bound for $\ell$: how is it derived? The authors should provide a detailed derivation.

17. Proof of Lemma 14, last formula, second inequality: why does it hold? The authors should provide a detailed derivation.

18. Proof of Lemma 14, last formula, last inequality: it is not a direct application of Freedman's inequality. The authors should provide a detailed derivation.

## Minor comments

1. Paragraph after the proof of Claim 5: "supported on its first $t - T^\ast$" --> "supported on its first $t - T^\ast - 1$"

2. Lemma 6: "algorithm 1" --> "Algorithm 1" (and everywhere else)

3. The end of Page 5, the third row of the derivation for $y_{t+1,j}$, the first case: one should not have $-1$ in the numerator.

4. Definition of $S$: I suggest to change the notation to $S_T$ since $S$ depends in $T$.

**Summary Of The Paper:**

This paper studies the convergence of the final iterate of (projected) SGD for (strongly) convex Lipschitz problems defined on bounded domain with almost surely bounded noise in the stochastic subgradients. In this setup, the authors derive the lower bounds for the last iterate of the projected sub-gradient descent that are valid for $T \geq d$, where $T$ is the number of iterations and $d$ is the dimensionality of the problem. The key idea behind proposed worst-case example is using the oracle that returns zero subgradient during the first $T-d$ iterations (the algorithm starts at the solution) and then the oracle returns "bad" subgradients that drift the method away from the solution. The derived lower bounds are valid for $1/\sqrt{T}$ and $1/\sqrt{t}$ stepsizes in the convex case and for $1/t$ stepsizes in the strongly convex case ($t$ is the iteration counter). Some upper bounds for SGD are obtained for $1$-dimensional problems that are "nearly linear".

**Summary Of The Review:**

Overall, the paper addresses a very interesting problem and the derived lower bounds can be seen as a solid contribution to the literature on SGD. However, some proofs (especially for the results from Section 4) contain a lot of unexplained parts and require a lot of polishing. This is the only reason why I give such a low score: in the current shape the paper cannot be accepted. However, if the authors fix all the raised issues, I will be happy to increase my score.

---

> ### Author Response · Authors · 2022-11-19
> **Response to Reviewer 6ixh**
>
> We thank you for your feedback!
>
> For weakness:
>
> Regarding bounded domain: the assumption of a bounded domain is standard in optimization. See, for example, Shamir and Zhang '2013 and Jain et al. '2019.
> We believe it is fair to make the assumption: considering the case where the domain is too large compared to the step size, it is impossible to get a good solution with only $T$ steps.
>
> Regarding bounded subgradients: we use bounded stochastic subgradients to ensure the concentration of the martingale difference.
> One can relax it to other assumptions, for example, being unbiased and sub-Gaussian, as long as one can still show concentration.
>
>
> For detailed comments:
>
> 1. We will add more comparisons with Jain et al. 2019 as suggested.
>
> 2. We agree this is an interesting problem, though "less non-adversarial" needs to be defined properly.
> Taking the suggestion "choose the subgradient somehow randomly to avoid bad behavior" as an example, it may be possible to transfer our current construction to build a lower bound for this case.
> Likely we can let the (random) oracle output 0 with probability 1-1/2(T-d) or something similar, which can lead to a similar effect of making the first T-d steps stay still.
>
> 3. We made some typos in the second equality. Let's derive it again here: by the definition of $z_{t,j}$ we know $z_{t,t-T^*}=0$. Thus the term in middle $\sum_{j=i-T^*}^{t-T^*}z_{t,j}(h_{t-T^*,j}-h_{i-T^*,j})$ is equal to $\sum_{j=i-T^*}^{t-T^*-1}z_{t,j}(h_{t-T^*,j}-h_{i-T^*,j})$. This term can be decomposed as $\sum_{j=i-T^*+1}^{t-T^*-1}z_{t,j}(h_{t-T^*,j}-h_{i-T^*,j})+z_{t,i-T^*}(h_{t-T^*,i-T^*}-h_{i-T^*,i-T^*})=\sum_{j=i-T^*+1}^{t-T^*-1}z_{t,j}a_j+z_{t,i-T^*}(a_{i-T^*}+1/2)>0$. We will add these details to the proof to make it clear.
>
> 4. We will add this as suggested.
>
> 5. These events are disjoint, so we can take equality.
>
> 6. We mean to rule out the possibility that $x_j>n_R$ and $x_{j+1}<n_L$.
> There is a flaw in our current parameters settings, which we think can be fixed immediately by modifying Definition 10 from $GD/\sqrt{T}$ to $4GD/\sqrt{T}$.
> Thanks for pointing it out, and we will clarify it.
>
> 7, 9. This constant $c\in(0,1]$ is the one in Definition 11 for the Nearly Linear Function. We will make it more clear.
> And as $E[y_j\mid F_{j-1}]\in[-\eta G,-c\eta G]$ (we made a typo in the draft by stating the interval $[-c\eta G,-\eta G]$), we know the upper bound of $\ell$.
>
> 8 and 15. Sorry for the confusion and flaws. We can upper bound $W_{(i,T+1]}$ by $(T-i)\eta^2G^2$ as a quick fix, which gives the tail probability $\Pr[\cdots\wedge t=i]$ on the page 9 by $\exp(\cdots-\Omega(c^2(T-i)))$ rather than $\exp(\cdots-\Omega(c(T-i)))$ as stated.
> This bound is still enough for the results of the near linear function. But it is unclear if it can recover the $O(\log T/\sqrt{T})$ for the general Lipschitz convex function. Thanks for pointing it out. We will clarify the proofs in the new version.
>
> 10. The space of event $t=i$ is contained in the space of event that $ W_{(i,T+1]}\le (T-i+1)(\eta G)^2$ (for the new upper bound) and $|\ell|\geq c\eta G (T+1-i)$.
>
> 11. We will fix the typos and make it easier to clarify.
> But we do not understand why you say we can not apply Freedman's inequality on Equation~(8).
>
> 12. We have for random variable $X\ge 0$, $E[X]=\int_{0}^{\infty}\Pr[X\ge t]d t$, which can be found in a probability textbook.
> We additionally use that if $X\in[0,GD]$, then $E[X]=\int_{0}^{GD}\Pr[X\ge t]d t$.
> The $O(\lambda)$ should be $O(1/\lambda)$ instead.
>
> 13. Thanks for pointing out this typo.
>
> The remaining comments for Lemma 14 seem similar to the ones for Theorem 15. We will address them as well in the new version.

---

> > ### Comment · Reviewer_6ixh · 2022-11-25
> > **Reply to Authors**
> >
> > I thank the authors for their detailed response and effort to improve the paper. I am increasing my score to 5. However, quite many details were changed in the proofs. I believe such changes require another full round of review.
> >
> > **Bounded domain assumption.** I agree that many works rely on the boundedness of the domain. However, for the in-expectation analysis of SGD it is not a necessity [1,2].
> >
> > **Re for detailed comments:**
> >
> > 1. Could the authors provide the comparison in the reply?
> >
> > 2. I encourage the authors to add this discussion to the paper.
> >
> > 3. The new derivation is correct.
> >
> > 5. I see, thank you for the clarification.
> >
> > 6. The authors should add the formal proof. In the current version, I do not understand why there exists solution $x^\ast$ such that $|n_R - x^\ast| \geq 4D/\sqrt{T}$ or $|n_L - x^\ast| \geq 4D/\sqrt{T}$.
> >
> > 7 and 9. I see, thank you for the clarification. Small remark: in the sentence "It is evident that $\ell \leq \ldots$", one should have $x_j > n_R$ in the end.
> >
> > 8, 11. Now I see how the authors apply Freedman's inequality (though the authors' response does not explain it): in (8) one can replace the right hand side with the maximum over deterministic quantity $|\ell|$ larger than $c\eta G (T+1 - i)$ and then apply Freedman's inequality. The change of the notation confused me, because previously $\ell$ was a stochastic variable.
> >
> > Regarding the comments about Lemma 14 (Lemma 15 in the new version): due to the time limitations I did not have chance to make another pass through the supplementary materials. Although the upper bound for the nearly linear functions is not a main contribution of the paper, multiple changes in this part require another thorough round of reviewing. That is why, I am slightly in favor of rejection.
> >
> > ---
> > References
> >
> > [1] Ghadimi, S., Lan, G., & Zhang, H. (2016). Mini-batch stochastic approximation methods for nonconvex stochastic composite optimization. Mathematical Programming, 155(1), 267-305.
> >
> > [2] Taylor, A., & Bach, F. (2019, June). Stochastic first-order methods: non-asymptotic and computer-aided analyses via potential functions. In Conference on Learning Theory (pp. 2934-2992). PMLR.

---

### Official Review · Reviewer_ao7o · 2022-10-27

**Confidence:** 3
**Correctness:** 3
**Technical Novelty And Significance:** 3
**Empirical Novelty And Significance:** Not applicable
**Recommendation:** 6

**Clarity, Quality, Novelty And Reproducibility:**

I found the paper well-written with sufficient details to explain the context and tools used. About quality and novelty -- although the authors build upon prior works on this topic, the proposed extensions are not incremental and require creative changes to yield improvements.
Reproducibility - Does not apply.

**Strength And Weaknesses:**

Strengths:


The authors make progress on a COLT 2020 published open problem, so the the question has certainly been deemed important by the community. In particular, even though SGD is a seemingly "simple" and widely-studied procedure, a tight characterization of the error of its final iterate is still not understood even in one dimension. Hence, the question is very natural. Both the results presented in the paper makes important contributions towards this problem.

The paper is well-written and presents the state of the problem well and give sufficient details of results and techniques of prior works. The proofs aren't long and complicated and it was nice that they could be presented in (almost) full details in the main text itself.

Weaknesses:

I think even though the results are important, in the bigger picture, the scope of the paper seems limited.
Firstly, the existing gap is only a $\log T$ factor.
More importantly, from an algorithm design perspective, we know (as discussed in the paper) that simple changes like returning the average iterate or last iterate but with a more complicated step-size suffices to get the optimal rate.
Therefore, the results may be (very) interesting to a very specific sub-community and not to the general ICLR audience.


I have some questions as well as minor corrections about the proof details:

- In the lower bound construction in Theorem 4, it seems that the constructed sub-gradient oracle maintains state -- in particular, on the same input, it gives different outputs as sub-gradients based on the iteration counter. This is a more powerful oracle than used in typical lower bound constructions that I am familiar with. I know that it is within the purview of settings in which the upper bounds are typically stated.
But is this something which has been used in prior works? Also, do the authors think that it may be possible to get rid of this additional power?

- Small missing argument before Lemma 6 -- for a complete argument, it seems that you should also argue that $H_i(z_t) < H_{t-T*}(z_t)$ for $i<t-T^*$ and then you have $I(x)\subseteq \\{ t-T^*,..d+1 \\}$  (it says $=$ currently), right?

- In the proof of Theorem 15, shouldn't it be $\eta G(T+1-i)\leq |l|\leq c\eta G(T+1-i)$ i.e. the lower bound does not have the factor $c$, since $\mathbb{E}[y_i] \in [-c\eta G,-\eta G]$? But it seems that this may affect the part of proof where you recover the $O(\log{T}/\sqrt{T})$ rate for general convex functions. What am I missing here?

- Typo: In the proof of Lemma 6, right hand side of third equality should have $a_j \frac{t-j-T^*}{\sqrt{T}}$

- Typo: In the proof of Theorem 15, shouldn't the definition of $W_{(i,T+1]}$ have $\tilde y_j^2$ as the summand?

- Typo: should be seminal instead of seminar in third line in page 2.

- Typo? I don't understand the sentence "Our results are in the strong versions regarding the third item" just before start of Section 3.

**Summary Of The Paper:**

The paper derives upper and lower bounds on the excess risk of last iterate of SGD in small dimensions with "standard" step-size schedules. They provide two main results: a lower bound of $\Omega\left(\frac{\log{d}}{\sqrt{T}}\right)$ on excess risk
for convex Lipschitz functions (which is also extended to strongly convex functions). Secondly, they define a sub-class which they call nearly-linear functions, and show that SGD with a step-size of $\frac{1}{\sqrt{T}}$ achieves the optimal rate of $O\left(\frac{1}{\sqrt{T}}\right)$ on this sub-class in one dimension.

**Summary Of The Review:**

I think the authors present important results on a fundamental problem. The only downside is that topic may be too specialized for ICLR.

---

> ### Author Response · Authors · 2022-11-19
> **Response to Reviewer ao7o**
>
> We thank you for your feedback! We agree that running average or more complicated step-size for last iterate are known to be optimal, but it is the standard last iterate that is most commonly used in practice. Our lower bound implies that averaging the solutions or so is better than the single last iterate with fixed step-size theoretically for the general dimension case.
>
> We answer the questions on proof details in order:
>
> Regarding the non-standard oracle, we can make the dependence on $t$ implicit as follows: in the first $T-d$ steps, the oracle doesn't stay still but moves very slowly so that $\|x_{T-d}\|_2$ is $1/\text{exp}(T)$. In this case, we can make the oracle give the same output for the same input independent of $t$, while the convergence rate is (approximately) preserved.
>
> Regarding missing argument before Lemma 6: in fact here we have $H_{t-T^*}(z_t)\ge H_{i-T^*}(z_t)$ for any $T^*+1\le i<t$, which is precisely the argument you wrote. We will make this point more clear.
>
> Regarding Theorem 15: recall that $\eta\in(0,1]$ and $-\eta G(T+1-i)\le \ell\le-c\eta G(T+1-i)$, so it is actually $c\eta G(T+1-i)\le |\ell|\le \eta G(T+1-i)$.
>
> Regarding Lemma 6: we will fix this typo, thank you for pointing it out!
>
> Regarding the definition in Theorem 15: we think the definition of $W_{(i,T+1]s}$ is consistent to the predictable quadratic variation process defined in Freedman's inequality (Theorem 12).
>
> Regarding other typos: we will change seminar to seminal, and by that sentence, we mean our lower bound is true even for sub-gradient descent, which one can access to a noiseless sub-gradient oracle. Our lower bound result for SGD is stronger in this sense.

---

### Official Review · Reviewer_F4uV · 2022-10-28

**Confidence:** 4
**Correctness:** 2
**Technical Novelty And Significance:** 2
**Empirical Novelty And Significance:** Not applicable
**Recommendation:** 5

**Clarity, Quality, Novelty And Reproducibility:**

Please see the comments on clarity and novelty above.

Other comments: The abstract should be one paragraph only, not two. Given that Theorem 4 is for a non-stochastic oracle while SGD is stochastic, the authors may want to add a discussion on that as well.

**Strength And Weaknesses:**

This paper provides the theoretical results on the upper bound and the lower bound for the final iterate of SGD. The authors consider dimension dependence on the convergence of SGD for non-smooth Lipschitz convex functions (upper bound) and nearly linear Lipschitz convex functions in one dimension (upper bound).

The first weakness is that the contribution of this paper is not really significant. There have been several lower bound results for SGD in the similar setting. The concept of nearly linear function is not motivated well enough. The contribution of the upper bound seems to be restrictive since it is dimension 1 and the results for SGD are not particularly interesting. In addition, the assumption that the domain is bounded is restrictive.

The second weakness is that some of the arguments are sloppy. For example:
- Too much use of asymptotic notations (e.g. Theorem 15) make it really hard to verify the proof. It would be much better if the authors could clarify this.
- Theorem 15, without loss of generality: please clarify this.
- Theorem 15, equation (10): how the conclusion changed into integration?



**Summary Of The Paper:**

This paper shows the lower bound for the final iterate of SGD and also upper bound for a class of nearly linear Lipschitz convex functions. The convergence result is dependent on the dimension $d$.

**Summary Of The Review:**

At the moment I do not support this paper because of the weaknesses noted above.

---

> ### Author Response · Authors · 2022-11-19
> **Response to Reviewer F4uV**
>
> We thank you for your feedback! Regarding significance: we believe this problem is important to the optimization community, as it is a formal open problem raised in COLT 2020, and our $\Omega(\log d / \sqrt{T})$ bound improves the previous best-known lower bound $\Omega(1 / \sqrt{T})$. In particular, the focus of the COLT open problem is on the $\log(d)$ term, which none of the previous works considered.
>
> The upper bound, even in one dimension, was asked in the COLT 2020 open problem as well. Thus, we believe it's of interest. The assumption of a bounded domain is standard in optimization.
>
> Regarding imprecise arguments: we apologize for the confusion caused and will improve the presentation.
>
> The case when $x_j<n_L$ can be handled with nearly the same arguments but only change the signs of some variables, say, e.g., $y_i$ and $\ell$.
> We feel it may be redundant to repeat the proofs.
>
> As for Equation~(10), in general we have for random variable $X\ge 0$, $E[X]=\int_{0}^{\infty}\Pr[X\ge t]d t$, which can be found in a probability textbook.
> We additionally use that if $X\in[0,GD]$, then $E[X]=\int_{0}^{GD}\Pr[X\ge t]d t$.

---

### Official Review · Reviewer_axZ4 · 2022-10-31

**Confidence:** 4
**Correctness:** 4
**Technical Novelty And Significance:** 2
**Empirical Novelty And Significance:** Not applicable
**Recommendation:** 3

**Clarity, Quality, Novelty And Reproducibility:**

The paper has some notation issues:
-- H_i is a function, and hence is not in R^d (you also don’t need this notation)
-- It should be explained that h_{i,j} is the j’th coordinate of h_i *before* h_{i,j} is defined. I would also change the notation to h^{(i)}_j
-- It should be clearly written that z_i are sub-gradient oracles (and again, before z_i is defined)
-- I didn’t understand what y_i is
-- It took me some time to find what D is in Definition 10. It also not explained what’s good about these points


**Strength And Weaknesses:**

I have the following concerns about the paper:
-- The main concern is that the result looks incremental. Basically, the idea is to freeze the iterate for the first T-d iterations, and then follow the proof which achieves Ω(log T / sqrt{T}) lower bound.
-- Related to the previous point, the sqrt{T} factor in the bound looks a bit arbitrary. I understand that the learning rate 1 / sqrt{T} (and other learning rates considered in hte paper) is chosen since it’s a popular choice for the optimization. However, it could as well be almost any other learning rate, giving a different lower bound. To clarify my point: when I started reading the proof, I was wondering why the bound was not Ω(log d / sqrt{d}), given that we simply ignore all but last d iterations. And the only reason for that turned out to be the choice of learning rate.


**Summary Of The Paper:**

The paper studies loss of the last iterate of the gradient descent in stochastic non-smooth settings. For the case d < T, they show the lower bound  Ω(log d / sqrt{T}) on the final iteration loss with the learning rate 1/\sqrt{T}, and show an upper bound O(1 / sqrt{T}) on the loss for certain class of functions.


**Summary Of The Review:**

Incremental work which would benefit from better presentation.

---

> ### Author Response · Authors · 2022-11-19
> **Response to Reviewer axZ4**
>
> We thank you for your effort! We believe your concern is a misunderstanding of our results.
>
> The final theoretical bounds unavoidably depend on the step size.
> The $\sqrt{T}$ factor is not arbitrary, but the (near) optimal theoretical choice for $T$-step SGD.
> As discussed in the intro, there is a non-standard step size to make the last iterate with convergence rate $O(1/\sqrt{T})$, but people do not know the rate for the fixed $O(1/\sqrt{T})$ step size, which motivates the COLT open problem.
> In this setting, previous work achieved the $\Omega(1 / \sqrt{T})$ lower bound, and our result indeed improves it to $\Omega(\log d / \sqrt{T})$ in a non-trivial way.
> Our lower bound implies that averaging the solutions or so is better than the single last iterate with fixed step size theoretically, which, unfortunately, is commonly used in practice.
> One can choose step-size $O(1/\sqrt{d})$ as you suggest, but the final rate will be worse than $1/\sqrt{T}$ by the optimal learning rate.
>
> Thanks for the good suggestions on the notation, and we will try to make the presentation clear.

---

### Author Response · Authors · 2022-11-19
**General Reply**

We thank all the reviewers for their time and insightful comments!
We want to emphasize the importance of our results, especially of the lower bounds, which demonstrate that taking the last iterate of SGD with standard step size is not as good as other strategies like taking the average or using non-standard step size in the worst case, which unfortunately is used a lot in practice.
We hope this can guide the implementation in reality and encourage comparing different step sizes and not using the last iterate directly.

We update the pdf version based on the reviews, with the main effort put into polishing the proofs.
Thanks for Reviewer 6ixh's comment on our mistake of upper-bounding the predictable quadratic variation process.
It is easy to fix for the upper bound of nearly linear
functions, as shown in the new version.
Nevertheless, it is still unclear whether the martingale-based argument can recover the $O(\log T/\sqrt{T})$ upper bound for the general Lipschitz function, which we do not treat as the main contribution. We are very sorry for the mistake.

There are some places the reviewers find confusing, which we explain in the rebuttal but do not clarify in the pdf.
If the reviewers still suggest we make the same clarifications in the pdf version, please feel free to reply to us, and we will be happy to make the change in the future version.

---

### Decision · Program_Chairs · 2023-01-20

**Decision:**

Reject

**Justification For Why Not Higher Score:**

- The main concern from the reviewers is that the results rely on the bounded domain assumption, which is quite restrictive.
- Many details were changed in the proofs during the revision. Therefore, the reviewers did not have change to check carefully and suggest the authors to resubmit to other venues for another review process.

**Justification For Why Not Lower Score:**

N/A

**Metareview: Summary, Strengths And Weaknesses:**

The paper shows the lower bound for the final iterate of SGD and also upper bound for a class of nearly linear Lipschitz convex functions. The convergence result is dependent on the dimension.

The reviewers think that the results are incremental and not significant enough for the publication at ICLR. The main concern from the reviewers is that the results rely on the bounded domain assumption, which is quite restrictive. Moreover, many details were changed in the proofs during the revision. Therefore, the reviewers did not have change to check carefully and suggest the authors to resubmit to other venues for another review process.

Please take the comments and suggestions from the reviewers in their detail reviews to improve the manuscript for the future submission.